# Transcriptomics of MASLD Pathobiology in African American Patients in the Washington DC Area [note 1]

**DOI:** 10.3390/ijms242316654

**Published:** 2023-11-23

**Authors:** Tanmoy Mondal, Coleman I. Smith, Christopher A. Loffredo, Ruth Quartey, Gemeyel Moses, Charles D. Howell, Brent Korba, Bernard Kwabi-Addo, Gail Nunlee-Bland, Leanna R. Rucker, Jheannelle Johnson, Somiranjan Ghosh

**Affiliations:** 1Department of Biology, Howard University, Washington, DC 20059, USA; tanmoy.mondal@howard.edu (T.M.); gmoses@uic.edu (G.M.); jheannellejohnson@gmail.com (J.J.); 2MedStar-Georgetown Transplantation Institute, Georgetown University School of Medicine, Washington, DC 20007, USA; coleman.i.smith@gunet.georgetown.edu; 3Department of Oncology, Georgetown University, Washington, DC 20007, USA; cal9@georgetown.edu; 4Department of Internal Medicine, College of Medicine, Howard University, Washington, DC 20007, USA; rquartey@howard.edu (R.Q.); chowell102@verizon.net (C.D.H.); 5Department of Microbiology & Immunology, Georgetown University, Washington, DC 20007, USA; brentkorba@gmail.com; 6Department of Biochemistry, College of Medicine, Howard University, Washington, DC 20059, USA; bkwabi-addo@howard.edu; 7Departments of Pediatrics and Child Health, College of Medicine, Howard University, Washington, DC 20059, USA; gnunlee-bland@howard.edu; 8Department of Internal Medicine, MedStar Georgetown University Hospital, Washington, DC 20007, USA; leanna.r.rucker@medstar.net

**Keywords:** MASLD, pathophysiology, transcriptomic analysis, African Americans, NAFLD

## Abstract

Metabolic-dysfunction-associated steatotic liver disease (MASLD) is becoming the most common chronic liver disease worldwide and is of concern among African Americans (AA) in the United States. This pilot study evaluated the differential gene expressions and identified the signature genes in the disease pathways of AA individuals with MASLD. Blood samples were obtained from MASLD patients (*n* = 23) and non-MASLD controls (*n* = 24) along with their sociodemographic and medical details. Whole-blood transcriptomic analysis was carried out using Affymetrix Clarion-S Assay. A validation study was performed utilizing TaqMan Arrays coupled with Ingenuity Pathway Analysis (IPA) to identify the major disease pathways. Out of 21,448 genes in total, 535 genes (2.5%) were significantly (*p* < 0.05) and differentially expressed when we compared the cases and controls. A significant overlap in the predominant differentially expressed genes and pathways identified in previous studies using hepatic tissue was observed. Of note, *TGFB1* and *E2F1* genes were upregulated, and *HMBS* was downregulated significantly. *Hepatic fibrosis signaling* is the top canonical pathway, and its corresponding biofunction contributes to the development of *hepatocellular carcinoma*. The findings address the knowledge gaps regarding how signature genes and functional pathways can be detected in blood samples (‘liquid biopsy’) in AA MASLD patients, demonstrating the potential of the blood samples as an alternative non-invasive source of material for future studies.

## 1. Introduction

Metabolic-dysfunction-associated steatotic liver disease (MASLD) (formerly referred to as non-alcoholic fatty liver disease (NAFLD)) is becoming the most common chronic liver disease worldwide [1]. The average global prevalence of MASLD in adults is 32%, although the prevalence rates vary by country and are higher in patients with metabolic syndromes, e.g., obesity, diabetes, hypertension, and hyperlipidemia [2,3]. MASLD can be classified as a progressive disease, potentially leading to metabolic-dysfunction-associated steatohepatitis (MASH), fibrosis, cirrhosis, and potentially hepatocellular carcinoma (HCC). Patients with MASLD are at increased risk of cardiovascular mortality [3,4]. The prevalence of MASLD in the overall United States (US) population has increased from ~20% in 2009 to 25–30% in 2023 [5,6,7]. This rise in prevalence has contributed to an increase in hospital admissions and liver transplant waitlists, resulting in an annual direct cost of USD 103 billion [8,9]. Liver biopsy has been the standard method of detection for assessing MASLD; however, it is not commonly performed due to its invasive nature, probability of sampling error, and high cost [10]. Liver biopsy is not recommended as a standard screening tool or as an initial diagnostic method in most patients [11].

The reported incidence of MASLD among African Americans (AAs) in the US is relatively low, but existing estimates may not be reliable due to AA being understudied as a subgroup in research focusing on chronic liver diseases [12]. AAs with chronic liver disease experience significant disparities in diagnosis and treatment due to systemic health inequities within the healthcare system [13]. These disparities result in differences in the diagnosis, management, and treatment of the disease, leading to worse outcomes for AAs compared to other racial groups [14].

Recent studies have suggested that gene expression studies using peripheral blood are a valuable source of biomarkers because of the depth of genetic information and the development of transcriptomic techniques [15,16,17,18,19]. Studies have assessed the peripheral blood transcriptome and its association with various diseases and drug responses [15,16,17]. The whole blood transcriptome has been evaluated in several studies for its diagnostic potential for early-stage cancer [18,19,20]. The primary objective of our study was to investigate the biological processes, signaling pathways, and key differentially expressed genes in peripheral blood samples during the early stages of development of MASLD in an AA population (an understudied population in the US). We also conducted a comparative analysis of our results and published datasets to identify pathways common across different ethnicities in blood samples. To our knowledge, global gene expression patterns in AA patients with MASLD, using blood as a primary source of bio-samples, have not yet been reported. In this study, we used an Affymetrix microarray to determine the expression profile of the whole blood transcriptome of patients with MASLD. 

## 2. Results

### 2.1. Participants’ Demography and Clinical Information

All recruited participants were from the Washington DC area. Table 1A displays the sociodemographic and lifestyle characteristics of all participants in each group. The mean age of the MASLD and control groups was 48.6 ± 7.5 and 42.6 ± 11.7 years, respectively (*p* = 0.10). A slightly higher, but insignificant, HbA1c level was observed in the MASLD group (6.5 ± 1.4%) compared to the control group (5.5 ± 0.2%, *p* = 0.78). Average BMI with the control group was nearly identical at 29.8 ± 6.8 and 30.1 ± 4.8 kg/m^2^, respectively. A significant number (*p* < 0.0001) of MASLD participants had hypertension as a comorbidity compared with the control group.

### 2.2. Global Gene Expression

A total of 21,448 transcripts were differentially expressed in the MASLD participants compared to the control group (false discovery rate (FDR) < 5%, *p*-value cutoff of <0.05). Out of these 21,448 genes, 535 genes were significantly and differentially expressed when we compared the cases and controls (fold change cutoff ±1.5 fold (or at least 1.5-fold) and *p*-value < 0.05)) (Figure 1) with 33% (174 genes) downregulated and 67% (361 genes) upregulated.

### 2.3. Top Biofunctions and Canonical Pathways Based on Global Expression Data

IPA was used to identify complex global expression data in the context of a biological system, such as functional roles, molecular processes, and key networks of the significantly differentially expressed genes in subjects with MASLD. The significant canonical pathways based on the overlap percentage and *p* < 0.05 were *coenzyme a biosynthesis*, *calcium transport I*, *TREM1 signaling*, *hepatic fibrosis signaling pathway*, and *estrogen receptor signaling,* with overlap percentages of 66.7%, 40.0%, 13.9%, 7.6%, and 7.6%, respectively (Table 2A). The top diseases and disorders based on the maximum number of genes and *p*-value < 0.05 were *organismal injury and abnormalities*, *immunological disease*, *inflammatory disease*, *connective tissue disorders*, and *inflammatory response*. The top molecular and cellular functions were *cell death and survival*, *cell function and maintenance*, *cell-to-cell signaling and interaction*, *cell morphology,* and *cellular compromise*. The top five physiological system development and functions were *organ development*, *nervous system development and function*, *digestive system development and function*, *hepatic system development and function, and behavior* (Table 2B).

The top network associated with the genes differentially expressed in the MASLD subjects in our IPA results contained 31 focus genes (Appendix A), with the highest score of 38 (*p*-score = −log10 *p*-value; a higher score indicates a stronger (significant) association of the gene with the network) and was associated with *hepatocellular carcinoma*, *proliferation of hepatic stellate cells*, *proliferation of liver cells*, and *Child–Pugh class A hepatocellular carcinoma*. Important canonical pathways linked to the top network were *hepatic cholestasis*, *hepatic fibrosis/hepatic stellate cell activation*, *hepatic fibrosis signaling* pathway, *Il-6 signaling*, *B cell receptor signaling*, *estrogen-mediated S-phase entry*, and *aryl hydrocarbon receptor signaling* (Figure 2). Most of the pathways and functions in this top network were significantly associated with the upregulation of *TGFB1* and downregulation of *E2F* and *E2F4*, which were identified as key genes in the network (Figure 2). However, our limited sample size for global gene expression data restricted us from conducting in-depth statistical analysis. 

### 2.4. Disease-Specific Gene Expression Validation by TLDA

For further validation, TLDA-based differential expression analyses were carried out to validate the global gene expression results on a specially designed human-liver-cancer-related gene set (Applied Biosystems, Catalog #4413255, Santa Clara, CA, USA) (Appendix A). The results indicated that the *hepatocellular carcinoma* and *hepatic fibrosis* category contained the largest number of gene networks and functions that were differentially expressed in the MASLD subjects. Of the 96 genes in the panel, 82 genes were differentially expressed in the MASLD subjects, of which 17 were upregulated and 65 were downregulated compared to the control group (Figure 3). 

### 2.5. Gene Network and Canonical Pathways (Validated Expression by TLDA)

The top canonical pathways associated with the genes differentially expressed in the MASLD subjects (based on the overlap percentage and *p*-value < 0.05) were the *role of tissue factor in cancer*, *chronic myeloid leukemia signaling*, *colorectal cancer metastasis signaling*, *molecular mechanisms of cancer*, and *hepatic fibrosis signaling* pathway (overlap percentages were 11.4, 10.7, 10.2, 8.1, and 7.0, respectively) (Table 3A). The top diseases and disorders (with a maximum number of molecules involved therein and *p*-value < 0.05) were *organismal injury and abnormalities*, *cancer*, *hematological disease*, and *tumor morphology*. The top molecular and cellular functions were *cell death and survival*, *cellular development, cellular growth and proliferation*, *cell function and maintenance*, *DNA replication*, *recombination, and repair*. The top five physiological system developments and functions were *tissue development*, *organismal survival*, *cardiovascular system development and function*, *organismal development*, and *connective tissue development and function* (Table 3B).

The TLDA result yielded a network of 22 genes (as supported by IPA knowledge base network analysis) that were associated with the key pathway of *hepatocellular carcinoma* (Appendix A). The construction of the network relied on information stored in the Ingenuity Pathways Knowledge Base (IPKB), along with actual expression data. Connections between differentially expressed genes were analyzed, focusing on those exhibiting a fold change of ≥1.5 and *p*-value < 0.05. The network highlighted canonical pathways, which were *hepatic cholestasis*, *hepatic fibrosis/hepatic stellate cell activation*, and the *hepatic fibrosis signaling pathway*. *TGFB1* was displayed as a central molecule as it is connected to most pathways and functions (Figure 4).

### 2.6. Comparison of TLDA and Global Expression

Thirteen genes were up- or downregulated in both the TLDA and global expression analyses in the MASLD subjects. The upregulated genes included *STAT3*, *ITGB1*, *AKT1*, *ADAM17*, *CFLAR*, *TGFB1*, *EP300*, *TCH4*, and *RUNX3*. The downregulated genes included *HMBS*, *E2F1*, *TP53*, and *SFRP2.* However, only *TGFB1* was significantly upregulated (*p* < 0.0004), and only *HMBS* (*p* < 0.01) and *E2F1* (*p* < 0.01) were significantly downregulated in the MASLD group compared with the controls (Figure 5). 

### 2.7. Comparison Analysis of Canonical Pathway Using IPA

Upon comparing our blood-based transcriptomic dataset with liver-tissue-based studies, we observed that the activation of canonical pathways was similar. The observed common canonical pathways were *molecular mechanisms of cancer*, *NAFLD signaling pathway*, *hepatic fibrosis signaling*, *IL-6 signaling*, and *estrogen receptor signaling* in the blood tissue (Figure 6). Additionally, we included a hierarchical clustering figure illustrating the common gene list observed through the comparison analysis (Figure 7). Ethnicity-based comparison analysis yielded three major pathways (*molecular mechanisms of cancer*, *hepatic fibrosis signaling*, *and estrogen receptor signaling*), similar to the MASLD global expression results (Figure 8).

## 3. Discussion

MASLD, the hepatic manifestation of metabolic syndrome, is the most common chronic liver disease in the United States, affecting approximately 100 million individuals [22]. Racial and ethnic disparities in the prevalence and severity of MASLD can be attributed to many factors, including genetics, environmental influences, and inequality in healthcare access [14]. These disparities significantly impact the diagnosis, management, and treatment of the disease, resulting in more severe outcomes for AAs than for other racial groups [13,14]. Early detection and effective management of liver disease are crucial for improving outcomes and preventing complications associated with MASLD.

In the process of global gene expression changes, the peripheral blood is one of the most useful tools for many disease biomarkers because of its circulating nature, which provides an opportunity to communicate and interact with diseased organs [20]. With the development of microarray technology, the transcriptome of peripheral whole blood can be readily profiled accurately and reproducibly. Due to the richness of gene expression information, the whole blood transcriptome is an attractive field of biomarker studies for various diseases, such as infectious diseases, neurodegenerative diseases, diabetes, and cancers [15,16,17,23,24,25,26,27]. The primary objective of our study was to use a less invasive way to conduct sample collection (‘liquid biopsy’) to investigate the biological processes, signaling pathways, and key differentially expressed genes associated with MASLD in an understudied AA population in the US. 

In our analysis of differentially expressed genes, we validated the differential expression of *STAT3*, *ITGB1*,* AKT1*, *ADAM17*, *CFLAR*, *TGFB1*, *EP300*, *TCH4*, *RUNX3*, *HMBS*,* E2F1*, *TP53*, and *SFRP2* in MASLD subjects. Of these genes, previous studies, including those that used liver tissue as source material, have reported that *STAT3*, *ADAM17*, *TGFB1*, *HMBS*, *E2F*, and *TP53* play significant roles in the *hepatic fibrosis signaling* pathway, and associated progression to *hepatocellular carcinoma* (HCC) is commonly differentially expressed [28,29,30,31,32,33]. Previous studies have reported that the activation of *STAT3* plays an important role in liver fibrosis, and the upregulation of *STAT3* and activation of *STAT* signaling pathways play a pro-inflammatory role during the pathogenesis of liver fibrosis [29]. Additionally, studies suggest that *STAT3* may contribute to hepatic fibrosis by interfering with the estrogen receptor signaling pathway [34]. The expression data show a significant activation of *STAT3*, correlating with the estrogen receptor signaling pathway identified through IPA analysis. Previous studies have indicated a higher incidence of MASLD in men, and in postmenopausal women (compared to premenopausal women), suggesting a potential association between estrogen and MASLD progression [35]. However, the specific mechanism behind estrogen’s role in MASLD remains unclear [35]. 

*ADAM17* is a disintegrin- and metalloproteinase-domain-containing protein that mediates the shedding of a wide variety of important regulators of inflammation, including cytokines and adhesion molecules [36]. A recent study suggested that cholestatic liver injury in humans is associated with increased hepatic *ADAM17* expression in liver tissue [36]. Consistent with this finding, we observed increased expression (upregulation) of *ADAM17* in our AA MASLD individuals.

In network analysis, the *hepatic fibrosis signaling* pathway was one of the most common and significantly enriched pathways in both array analyses. The upregulation of *TGF-β1* and its association with the *hepatic fibrosis signaling* pathway is one of the most important factors in the pathogenesis of liver fibrosis [37]. We observed that *TGF-β1* was significantly upregulated in all MASLD participants compared to controls. *TGF-β1* signaling mechanisms play a key role in maintaining normal homeostasis in the liver [38]. Upregulation of *TGF-β1* plays a crucial role in the activation of hepatic stellate cells (HSCs) and the generation of the extracellular matrix after acute and chronic liver damage, which accelerates the development of MASLD [39]. We also observed that *TGF-β1* stands as a “signature gene” through our network analysis that connects the maximum number of pathways and functions in the network with *TGF-β1* activation (Appendix A).

Several lines of evidence have shown that cell cycle regulatory proteins can also modulate metabolic processes [31,40]. Transcription factor *E2F1* is a central player in cell cycle progression, DNA damage response, and apoptosis [31]. Recent findings support cell-cycle-independent roles of *E2F1* in various tissues that contribute to global metabolic homeostasis [31]. We observed a significant downregulation of *E2F1* in all MASLD patients compared to that in the control population. 

*HMBS* expression was significantly downregulated in our MASLD study population. *HMBS* is a novel metabolic tumor suppressor gene in HCC [30]. Reported studies suggest that *HMBS* inactivation induces accumulation of its toxic substrate, porphobilinogen, and confers a high risk of HCC development [30]. It is known that abnormal amounts of liver fat in MASLD patients cause inflammation by invading immune cells and secreting cytokines, one of which is interleukin-6 (*IL-6*), a crucial immunomodulatory cytokine that influences the pathophysiology of MASLD [41]. In addition to observing significant downregulation of *E2F1* and *HMBS*, we also identified a notable decrease in the expression of the *TP53* gene in the majority of MASLD subjects (*n* = 11). *TP53* is known for its role in various signaling pathways that induce apoptosis [42]. The *TP53* gene is known for its involvement in numerous signaling pathways that trigger apoptosis [42]. In relation to MASLD, studies have uncovered the therapeutic implications of *TP53* regulation [41]. However, the role of *TP53* in MASLD remains controversial, with some studies proposing that activated *TP53* contributes to MASLD pathogenesis, while others argue that suppressing *TP53* activation worsens liver steatosis [32].

Our IPA-based analysis of cellular processes and pathways revealed the dysregulation of sets of genes associated with specific pathways in MASLD subjects: *proliferation of hepatic stellate cells*, *proliferation of liver cells*, and *Child–Pugh class A hepatocellular carcinoma*. These pathways are also associated with an increased risk of future liver diseases, including HCC [43,44]. Previous studies have reported that the advancement of fibrosis in MASLD is primarily driven by the stimulation of hepatic stellate cells [43]. These activated cells undergo a transformation into myofibroblasts, which display increased proliferation rates and contribute to the production of an extracellular matrix that is notably influenced by *TGF-β1* [45]. These activated stellate cells can then be transformed into myofibroblast-like cells to promote fibrosis in response to liver injury or chronic inflammation, leading to liver cirrhosis and ultimately cancer [43]. Even though the participants included in our study were not currently diagnosed with liver cancer, our blood-based transcriptomes suggest potential future risk among MASLD patients regarding development of liver cancer.

IPA analysis also provided us with an opportunity to conduct a comparative study using a larger dataset from a public repository database. We identified GEO datasets for comparing liver tissue and our blood specimens and comparing different ethnic and racial groups [46,47,48,49,50,51]. We noticed that there were more similarities than differences. However, *SPP1* expression was strongly linked to tumor progression and liver cancer in liver tissue from Caucasian patients with liver cancer (HCC) [50]. In other studies, *PNPLA3* and *TM6SF2* variants were noted as potential risk factors for MASLD, and it was observed that AA had a lower frequency of the *PNPLA3* variant compared to other ethnic groups [52]. The comparative study yielded common canonical pathways in both liver tissue and whole-blood-based studies, such as *molecular mechanisms of cancer*, *NAFLD signaling pathway*, *hepatic fibrosis signaling*, *IL-6 signaling*, and *estrogen receptor signaling* (Figure 6), indicating the utility of using blood samples for MASLD-associated transcription studies [15,16,17,18,19,20]. In addition, we observed that the activation of *molecular mechanisms of cancer*, *hepatic fibrosis signaling*, and *estrogen receptor signaling* was common across all ethnicities (Figure 8).

The use of peripheral blood (“liquid biopsy”) for MASLD transcriptome studies offers significant advantages over tissue biopsies, considering its cost, acceptability, and less-invasive nature. This study demonstrates the feasibility of using peripheral blood to provide genetic information that is consistent with the genetics of liver tissue biopsies.

## 4. Materials and Methods

### 4.1. Study Ethics, Participants, and Selection of Participants

The study population consisted of 39 adults, including both male and female participants, who self-identified as AA and were born in the US. All participants responded to an advertisement through Howard University and Georgetown University Community Newsletter via email and/or flyers and public announcements and were recruited with their informed consent. The protocol was approved by Georgetown-MedStar IRB (MODCR00002260). The participants were separated into two groups: the control group, which included individuals without MASLD, and the patient group, which included those with the disease. Participants with MASLD (*n* = 23; male = 11, female = 12) were recruited from the MedStar Georgetown Transplant Institute. We selected only those patients who had confirmed hepatic steatosis (based on their imaging/biopsy records supported by the presence of hepatic steatosis on cross-sectional imaging, liver elastography, and/or histological confirmation by percutaneous liver biopsy) and exhibited one or more comorbid metabolic features, viz., type 2 diabetes, hypertension, hyperlipidemia, or obesity (Appendix A). Individuals with progression to severe fibrosis or cirrhosis were not included because of the small size of the participant groups and the wide spectrum of tissue features present during different stages of liver disease; we chose to limit subject inclusion to earlier stages of liver disease to increase the homogeneity of disease presentation in the different subjects. Patients with other potential causes of liver disease, including viral, immunological, iron storage disease, Wilson disease, or alpha 1 antitrypsin deficiency, were excluded from the study. Participants with heavy alcohol use were also excluded from the study. We also recruited participants without reported MASLD (*n* = 16; male = 9, female = 7) from the same geographical region (Howard University, Washington, DC, USA) as a comparator (control) group. These individuals were negative for HCV and HBV and had normal liver enzyme profiles.

A questionnaire was provided to all participants to collect information about smoking history, alcohol consumption, and comorbidities, such as BMI, hypertension, and cardiovascular disease (Table 1A). For each subject, a retrospective chart review was performed to identify the incidence of comorbid metabolic conditions, including obesity (classified by body mass index (BMI), type 2 diabetes mellitus, hypertension, and dyslipidemia). Additionally, liver enzyme levels and alcohol use history were reviewed to exclude those with significant liver injury or suspected alcohol-associated liver disease, respectively.

### 4.2. RNA Isolation & cDNA Synthesis

Whole blood was collected in a DNA/RNA Shield™ Blood Collection Tube (Manufacturer: Zymo Research, Irvine, CA, USA, Cat # R1150) during recruitment by experienced phlebotomists. Blood collection tubes were prefilled with 6 mL DNA/RNA Shield™ for direct collection of up to 3 mL whole human blood. DNA/RNA Shield lyses cells, inactivates nucleases and infectious agents (e.g., viruses and pathogens), and is ideal for safe sample storage and transport at ambient temperatures. RNA was extracted from DNA/RNA Shield tubes using the Quick-DNA/RNA™ Blood Tube Kit (Cat. # R1151, Zymo Research, Irvine, CA, USA) according to the manufacturer’s instructions. DNA contamination was removed using an Applied Biosystems Inc. (ABI) DNA-free kit (Cat # AM 1906, ThermoFisher, Waltham, CA, USA). RNA was quantified using a NanoDrop™ One spectrophotometer (Thermo Fisher Scientific, Wilmington, DE, USA). The ratio of absorbance at 260 and 280 nm was used to assess the purity of DNA and RNA. Total RNA was reverse-transcribed to cDNA using a High-Capacity cDNA Reverse Transcription Kit (Part # 4368814; Applied Biosystems, Waltham, MA, USA) according to the manufacturer’s instructions. The reaction mixture (20 μL total volume) was incubated at 25 °C for 10 min, and then at 37 °C for 120 min. Finally, the mixture was heated at 85 °C for 5 min. The cDNAs were stored at −15 to −25 °C if not used immediately (within 24 h) or stored at 2–6 °C.

### 4.3. Microarrays and Global Gene Expression

To assess the overall transcriptome in the disease process, we conducted a global gene expression array analysis in a small subgroup of eight participants selected from the total study population, consisting of four individuals with MASLD and four individuals in the control group. The decision to work with a small subgroup was driven by the difficulty in identifying multiple participants with closely aligned experimental characteristics within such a compact cohort during the recruitment phase. In order to assess the initial screening of genes showing differential expression, we carefully selected the best-matched combination of participants, ensuring that variables such as age, sex, BMI, HbA1c, and other significant demographic and clinical factors were balanced across both groups (Table 1B for details). Microarray analysis was performed using the Clarion S Assay, Affymetrix (Cat. # 902927, Santa Clara, CA, USA), with 24,000 transcripts. Differentially expressed gene sets were analyzed from the microarray results using a one-way ANOVA model by Partek Genomic Suites (GS- V.4.1; Partek Inc., St. Louis, MI, USA). Probe summarization and probe set normalization were performed using the GC-RMA algorithm, which included GC-RMA background correction, quantile normalization, log2 transformation, and median polish probe set summarization. The study outcomes maintained a false positive percentage <5% and a significance level. 

### 4.4. High-Throughput TaqMan^®^ Low-Density Array (TLDA) Gene Expression with Liver Cancer Profiler Array

In the TLDA study, 31 participants were included, with 19 and 12 participants in the MASLD and control groups, respectively (Table 1B). In this process, differential gene expression validation was performed using TaqMan^®^ Array (FAST Plate) for TaqMan^®^ Array Human Liver Cancer 96-well plate, fast (configurable) (Applied Biosystems, Catalog #4413255, Santa Clara, CA, USA) on the ABI QuantStudio 5 Real-Time PCR System (Thermo Fisher Scientific, Wilmington, DE, USA) for quantification. In this TaqMan^®^ Array, the expression of important human-liver-cancer-related genes in patients with MASLD was compared with that in controls. The listed genes (*n* = 96) of the array cards were based on key genes involved in the progression of hepatic fibrosis, hepatocellular carcinoma (HCC), proliferation of hepatic stellate cells, and other forms of hepatocarcinogenesis. Fatty-liver-disease-associated genes commonly alter signal transduction pathways, as well as those involved in other dysregulated biological pathways such as hepatic cholestasis, cell cycle, apoptosis, and inflammation.

### 4.5. Identification of Cellular Processes, Biofunctions, and Canonical Pathways by Ingenuity Pathways Analysis (IPA)

From the differential gene expression datasets described above, the identification of cellular processes and pathways by IPA was performed according to the method described in our earlier study. Briefly, datasets comprising gene identifiers and corresponding expression values (fold change) from the microarray experiment were imported into IPA. Differentially expressed gene identifiers were mapped to related changes in biofunctions. The networks were generated algorithmically based on their connectivity. Using IPA, we identified the top network by amalgamating a large set of differentially expressed genes with the goal of uncovering the most extensive array of relationships among the focus genes [53]. A score (*p*-score = −log10 (*p*-value)) according to the fit of the set of supplied genes and a list of biological functions stored in the Ingenuity Knowledge Base are generated [53]. Networks were “named” on the most prevalent functional group(s) present. Canonical pathway (CP) analysis identified function-specific genes that were significantly present within the networks. 

The IPA was also used to examine and compare the relationship between enrichment results from differentially expressed genes from our experiment and the curated datasets from GEO (Gene Expression Omnibus, NCBI, https://www.ncbi.nlm.nih.gov/geo/ accessed on 26 April 2023) [54]. We identified two GEO datasets (GSE135251 and GSE126848) from the NCBI data bank [46,47,48]. Both the datasets involved the characterization of transcriptional changes in liver disease progression, with liver biopsies tissue samples with more than 220 NAFLD cases at different fibrosis stages and 25 controls, all of which underwent high-throughput RNA sequencing [46,47,48]. We also performed comparison analysis based on the ethnicity differences. In this analysis, we selected Caucasian (GSE151530), Hispanic (GSE115469), and Chinese (HCL_Liver.GPL20795) populations from the GEO data repository to compare with our AA expression data [49,50,51]. The comparison analysis function in IPA was employed to assess the similarity and difference among the enriched pathways and upstream regulators. A heatmap was generated to visualize the canonical pathways and upstream regulators that correlate with the original experimental dataset. The analysis will help us identify pathways and upstream regulators whose activity may increase or decrease based on activation z-scores. Activation z-scores of ≥2 or ≤−2 are considered significant, and they are used to determine the activation states (“increased” or “decreased”) of implicated biological functions.

### 4.6. Statistical Analysis

Statistical analysis was performed using the chi-squared test and *t*-test for comparisons between the MASLD and control groups to observe any significant differences in the clinical parameters and comorbidities. Reported data are represented as mean ± SEM, and figures were produced using GraphPad Prism (version 8) software. Differences were considered statistically significant at *p* < 0.05.

## 5. Conclusions

Our pilot study focused on identifying key differentially expressed genes and their associated canonical pathways in AA patients with MASLD using peripheral blood samples. Our results begin to fill knowledge gaps on how the genes and functional pathways identified from liquid biopsy are dysregulated in AA patients with early-stage MASLD. The study reveals that *TGFB1*, *E2F1*, and *HMBS* could potentially serve as markers to identify the activation of the hepatic fibrosis signaling pathway and its corresponding biological function in the development of hepatocellular carcinoma. The correlation study also provides information about similarities with hepatic tissue from NAFLD patients, even across different ethnicities, offering a broad perspective on the future risk of developing cancer. However, multiethnic large-scale population validation studies are required to support these early findings and to discover and validate potential blood-based MASLD biomarkers.

## Figures and Tables

**Figure 1 ijms-24-16654-f001:**
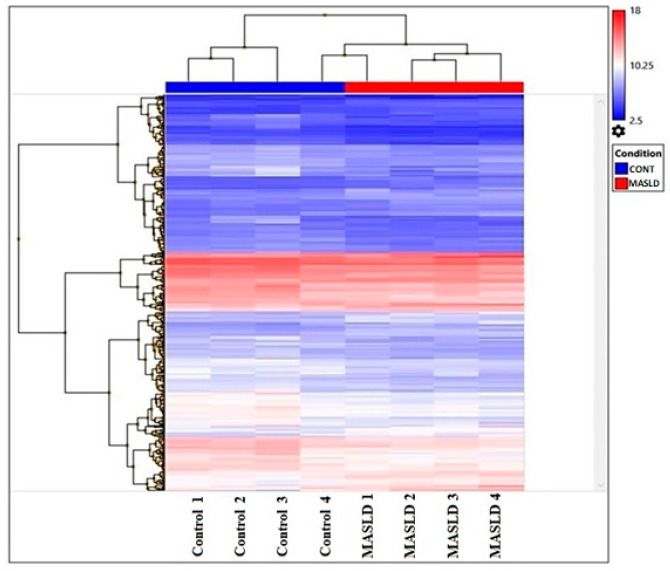
Heatmap showing the hierarchical clustering of 535 genes differentially expressed (361 were upregulated (red) and 174 were downregulated (blue), adjusted *p* < 0.05) between the control and MASLD groups. The dendrogram displays the clustering genes, where closely related genes were grouped together. In the figure, the control group and MASLD group are labeled at the top with blue and red colors, respectively.

**Figure 2 ijms-24-16654-f002:**
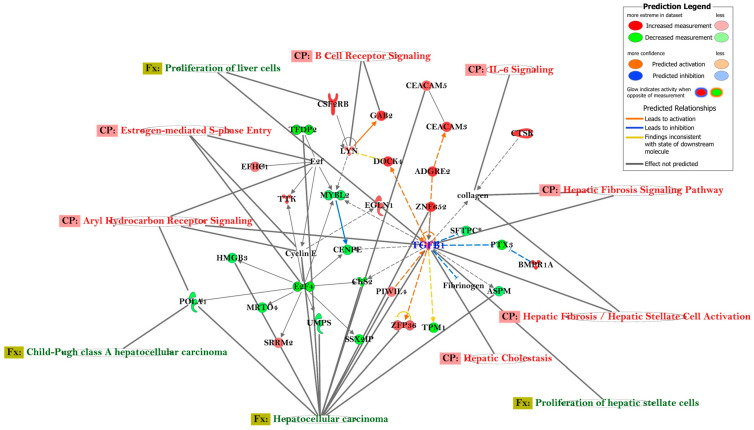
Network of differentially expressed genes in the global gene expression data in the MASLD participants, relative to control subjects. The network was generated using Ingenuity Pathway Analysis (IPA) from QIAGEN, USA, March 2023 release, and incorporated the relative gene expression levels measured in the patients’ group [21]. The construction of the network relied on information stored in the Ingenuity Pathways Knowledge Base (*IPKB*), along with actual expression data. Connections between differentially expressed genes were analyzed, focusing on those exhibiting a fold change of ≥1.5 and *p*-value < 0.05. The network includes canonical pathways (CP) associated with the analyzed genes, which were represented within an oval-shaped circle. Genes that are upregulated are represented by geometric figures in red, while downregulated genes are depicted in green. The intensity of the red and green colors reflects the degree of up- or downregulation, respectively, in the expression dataset. Gray-filled color identifiers represent molecules that either did not meet the analysis cutoff (*p* < 0.05) in the dataset or only contain identifiers without any associated expression values.

**Figure 3 ijms-24-16654-f003:**
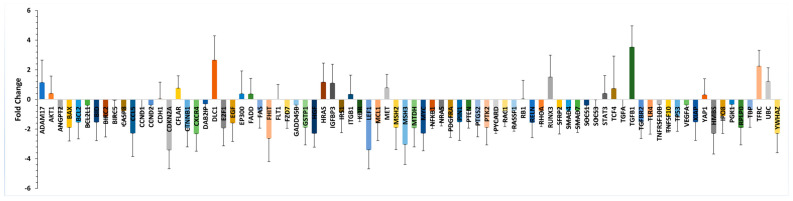
Relative expression levels of genes were differentially expressed in MASLD subjects. Genes in MASLD subjects that were differentially expressed (*p* < 0.05) in the TLDA analysis are listed. Gene expression levels for MASLD subjects are displayed relative to the mean levels (fold change) for each gene in the control group.

**Figure 4 ijms-24-16654-f004:**
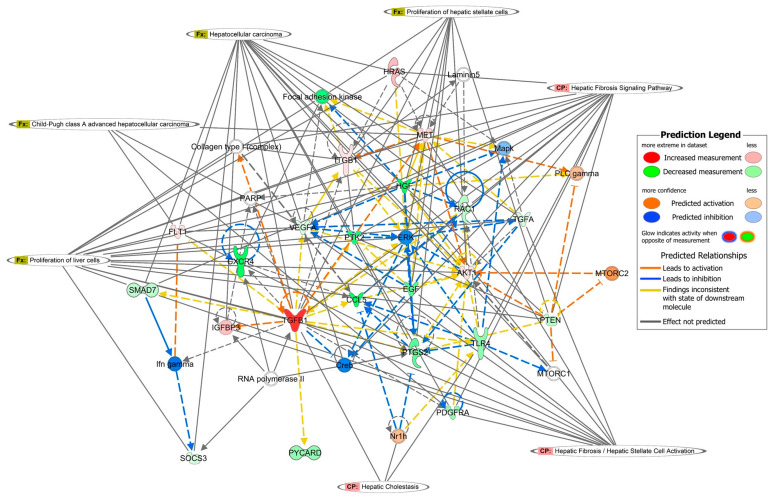
Network of differentially expressed genes in the TLDA gene expression data in the MASLD participants, relative to control subjects. The network was generated using Ingenuity Pathway Analysis (IPA) from QIAGEN, Germany, March 2023 Release, and incorporated the relative gene expression levels measured in the MASLD group [21]. The construction of the network relied on information stored in the Ingenuity Pathways Knowledge Base (IPKB), along with actual expression data. Connections between differentially expressed genes were analyzed, focusing on those exhibiting a fold change of ≥1.5 and *p*-value < 0.05. The network includes canonical pathways (CP) associated with the analyzed genes, which are represented within an oval-shaped circle. Genes that are upregulated are represented by geometric figures in red, while downregulated genes are depicted in green. The intensity of the red and green colors reflects the degree of up- or downregulation, respectively, in the expression dataset. Gray-filled color identifiers represent molecules that either did not meet the analysis cutoff (*p* < 0.05) in the dataset or only contain identifiers without any associated expression values.

**Figure 5 ijms-24-16654-f005:**
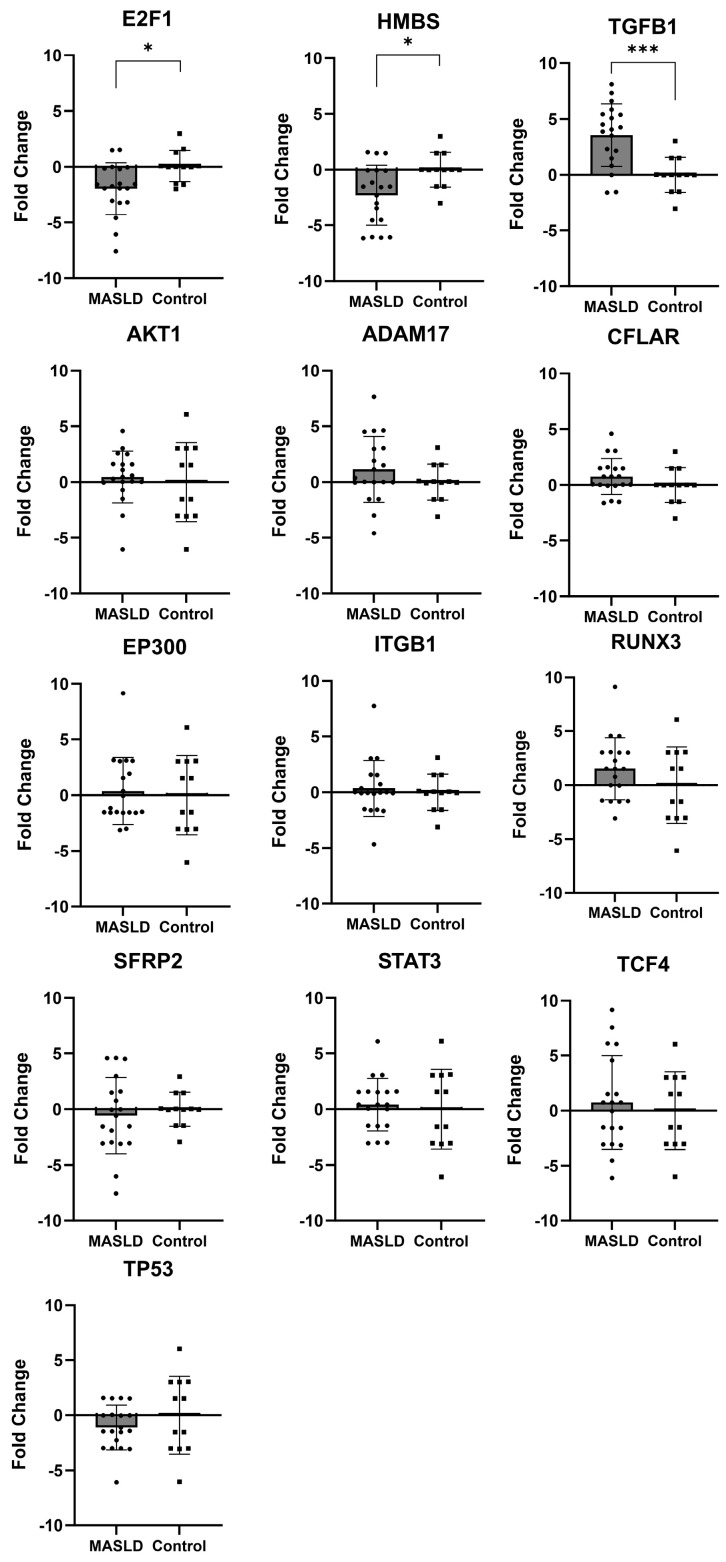
Relative expression levels of genes were differentially regulated in MASLD subjects in both the TLDA and global expression analyses. Gene expression levels are displayed relative to the mean levels for each gene (fold change) in the control group. Statistical analysis was performed by *t*-test (* *p* < 0.01 and *** *p* < 0.0001).

**Figure 6 ijms-24-16654-f006:**
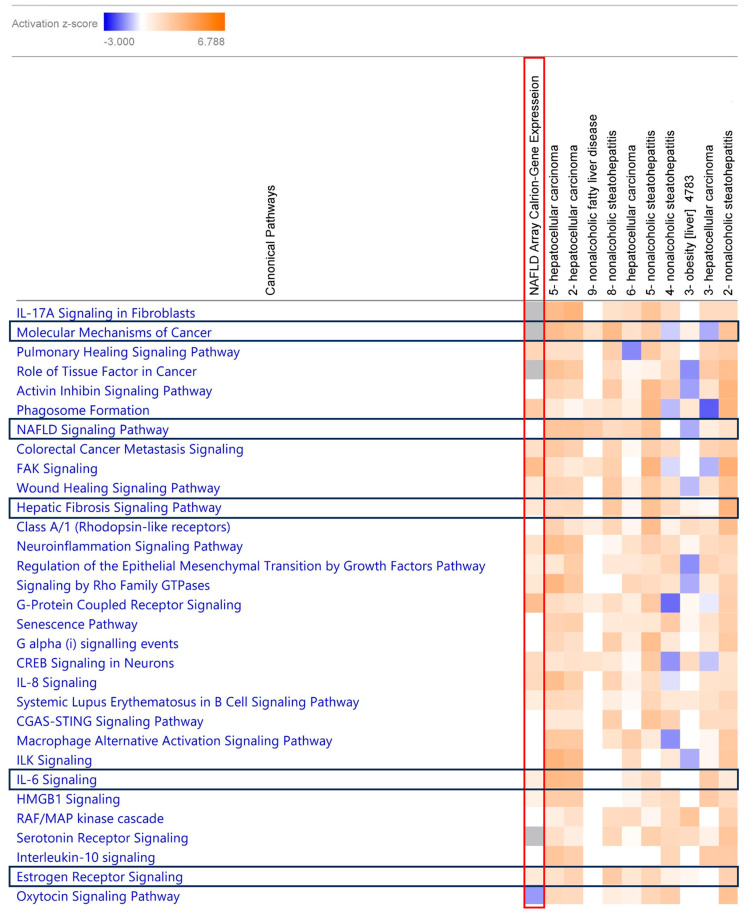
Comparison analysis between global transcriptomic data from blood sample and GEO curated NAFLD liver tissue sample. The heatmap of the canonical pathway across different analyses displaying the z-score from pathway activation analysis. Orange and blue rectangles represent activation and inhibition, respectively. The red-box column is our analyzed dataset from the global expression data and black-box rows represent identified similar canonical pathways with the global data. Activation z-score was used to infer the activation states (“increased” or “decreased”) of identified biological functions. Z-scores of ≥2 or ≤−2 are considered significant.

**Figure 7 ijms-24-16654-f007:**
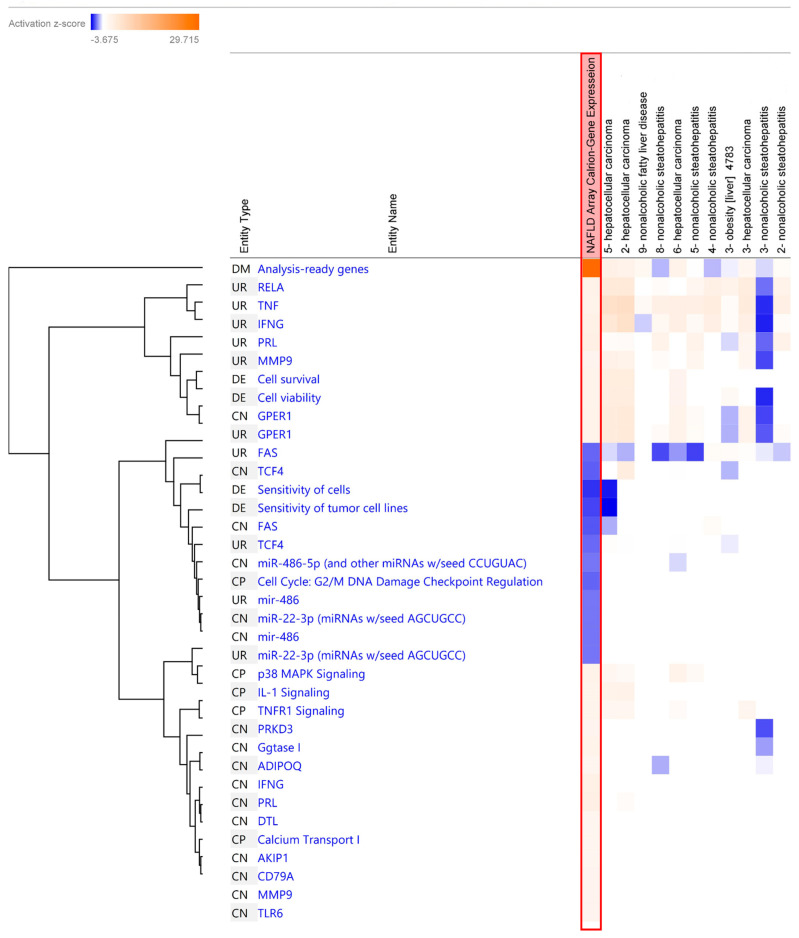
Hierarchical clustering of common genes based on comparison analysis between experimental dataset and GEO curated dataset. The heatmap of the specific genes across different analyses displaying the z-score-based orange and blue rectangles, which represent activation and inhibition, respectively. The red-box column is our analyzed dataset from the global expression data. Activation z-score was used to infer the activation states (“increased” or “decreased”) of identified genes. Z-scores of ≥2 or ≤−2 are considered significant.

**Figure 8 ijms-24-16654-f008:**
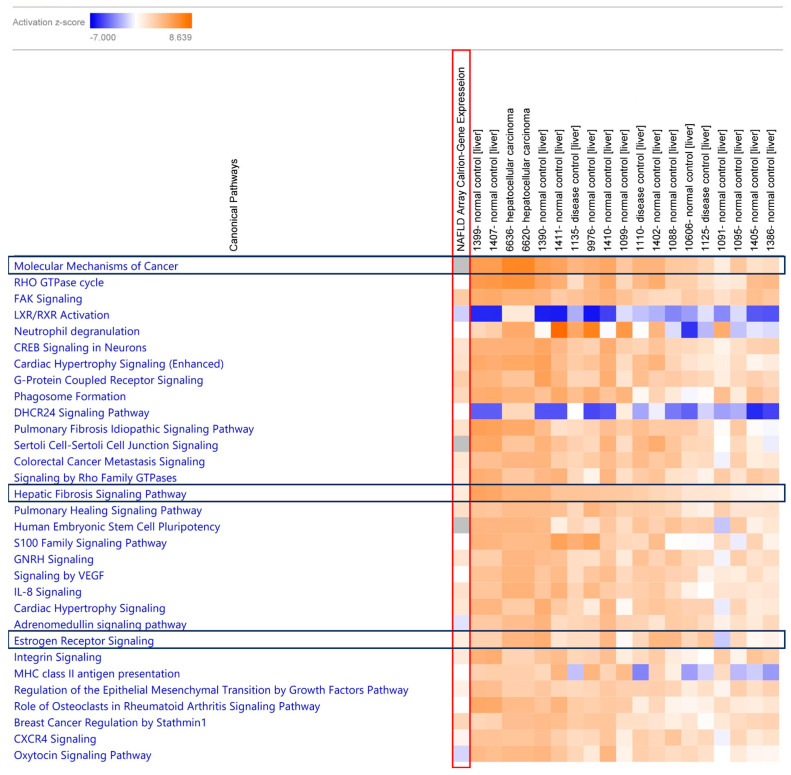
Canonical pathway and comparison analysis between experimental dataset and GEO curated dataset. The heatmap of the canonical pathway across different analyses based on the ethnicity differences columns displaying the z-score from pathway activation analysis. Orange and blue rectangles represent activation and inhibition, respectively. The red-box column is our analyzed dataset from the global expression data and black-box row represents identified similar canonical pathways with the global data. Activation z-score was used to infer the activation states (“increased” or “decreased”) of identified biological functions. Z-scores of ≥2 or ≤−2 are considered significant.

**Table 1 ijms-24-16654-t001:** (**A**) Study participants’ demography. (**B**) Study participants’ demography based on applied array.

**(A)**					
**Characteristics**	**Primary Cohort for Global Gene Expression Study (*n* = 39)**				
	**MASLD Participants (*n* = 23)**		**Non-MASLD** **Controls * (*n* = 16)**		** *p* ** **-Value**
**Age (Y)**	48.63 ± 7.51		42.60 ± 11.70		0.10
**Gender (M/F)**	11/12		9/7		-
**BMI (kg/m^2^)**	29.75.2 ± 6.82		30.14 ± 4.77		0.78
**HbA1c (%)**	6.35 ± 1.22		5.37 ± 0.17		0.32
**Hypertension (n)**	13		4		<0.0001
**Had Stroke (n)**	4		0		-
**Infrequent Alcohol Consumption (n) ****	20		10		0.054
**Smoking (n)**	8		4		0.75
**(B)**					
**Characteristics**	**Primary Cohort for Global Gene Expression Study (*n* = 8)**		**Validation Cohort for TLDA Study (*n* = 31)**		
	**MASLD Participants (*n* = 4)**	**Non-MASLD Control (*n* = 4)**	**MASLD Participants (*n* = 19)**	**Non-MASLD Control * (*n* = 12)**	** *p* ** **-Value (Validation Cohort)**
**Age (Y)**	49.00 ± 3.5	49.75 ± 8.89	48.63 ± 8.1	35.9 ± 11.1	0.0029
**Gender (M/F)**	2/2	2/2	9/10	7/5	-
**BMI (kg/m^2^)**	32.20 ± 5.4	30.92 ± 4.2	29.23 ± 7.1	30.1 ± 5.5	0.73
**HbA1c (%)**	5.55 ± 0.4	5.3 ± 0.1	6.48 ± 1.4	5.45 ± 0.2	0.36
**Hypertension (n)**	2	2	11	2	<0.0001
**Had Stroke (n)**	1	0	3	0	-
**Infrequent Alcohol Consumption (n) ****	3	3	17	7	0.056
**Smoking (n)**	2	3	6	1	0.86

* Participants with no major liver problem or MASLD recorded in their medical records and those who have tested negative for the HEP-C virus and have no previous records of viral infection (Tested). ** Once in a week or less.

**Table 2 ijms-24-16654-t002:** (**A**) IPA analysis based on global gene expression results: Top canonical pathways with their corresponding *p*-value in the MASLD population (*n* = 8). (**B**) IPA analysis based on global gene expression results: Top Diseases and Bio functions associated with dysregulated genes in MASLD participants.

(**A**)		
**Top Canonical Pathways ***	** *p* ** **-Value**	
** *Coenzyme A Biosynthesis* **	6.27 × 10^−3^	
** *Calcium Transport I* **	7.75 × 10^−4^	
** *TREM1 Signaling* **	1.75 × 10^−3^	
** *Hepatic Fibrosis Signaling Pathway* **	5.09 × 10^−3^	
** *Estrogen Receptor Signaling* **	6.25 × 10^−3^	
**(B)**		
**Top Diseases and Bio Functions ***		

**Diseases and Disorders**	# Molecules	*p*-Value Range
** *Organismal Injury and Abnormalities* **	829	2.85 × 10^−2^–6.03 × 10^−6^
** *Immunological Disease* **	206	2.85 × 10^−2^–6.03 × 10^−6^
** *Inflammatory Disease* **	160	2.85 × 10^−2^–6.03 × 10^−6^
** *Connective Tissue Disorders* **	154	2.85 × 10^−2^–6.03 × 10^−6^
** *Inflammatory Response* **	146	2.70 × 10^−2^–6.03 × 10^−6^

**Molecular and Cellular Functions**	# Molecules	*p*-value Range
** *Cell Death and Survival* **	181	2.85 × 10^−2^–6.29 × 10^−4^
** *Cell Function and Maintenance* **	53	2.85 × 10^−2^–3.21 × 10^−4^
** *Cell-To-Cell Signaling and Interaction* **	53	2.85 × 10^−2^–7.75 × 10^−4^
** *Cell Morphology* **	34	2.85 × 10^−2^–8.15 × 10^−4^
** *Cellular Compromise* **	15	2.85 × 10^−2^–7.75 × 10^−4^

**Physiological System Development and Function**	# Molecules	*p*-value Range
** *Organ Development* **	18	2.48 × 10^−2^–3.63 × 10^−4^
** *Nervous System Development and Function* **	15	2.25 × 10^−2^–8.15 × 10^−4^
** *Digestive System Development and Function* **	10	2.02 × 10^−2^–3.63 × 10^−4^
** *Hepatic System Development and Function* **	10	2.02 × 10^−2^–3.63 × 10^−4^
** *Behavior* **	3	2.16 × 10^−2^–2.16 × 10^−3^

* The data were generated using Ingenuity Pathway Analysis (IPA) from QIAGEN, USA, March 2023 release, and incorporated the relative gene expression levels measured in the patients’ group [21]. The *p*-value was measured to understand the likelihood that the association between a set of molecules in our dataset and a related disease or function is due to random association. The smaller the *p*-value (which means a larger −log of that value), the less likely that the association is random and the more significant the association. In general, *p*-values ≤ 0.05 (−log = 1.3) indicate a statistically significant, non-random association. The *p*-value was calculated using a right-tailed Fisher’s exact test [21]. # Molecules: number of significant molecules that are associated with each function.

**Table 3 ijms-24-16654-t003:** (**A**) IPA analysis based on TLDA results: top canonical pathways associated with dysregulated genes in MASLD subjects. (**B**) IPA analysis based on TLDA results: top diseases and biofunctions with dysregulated genes in MASLD subjects.

**(A)**		
**Top Canonical Pathways ***		** *p* ** **-Value**
** *Role of Tissue Factor in Cancer* **		7.24 × 10^−28^
** *Chronic Myeloid Leukemia Signaling* **		1.87 × 10^−34^
** *Colorectal Cancer Metastasis Signaling* **		2.02 × 10^−31^
** *Molecular Mechanisms of Cancer* **		8.95 × 10^−39^
** *Hepatic Fibrosis Signaling Pathway* **		5.16 × 10^−29^
**(B)**		
**Top Diseases and Bio Functions ***		

**Diseases and Disorders**	# Molecules	*p*-Value Range
** *Organismal Injury and Abnormalities* **	82	1.52 × 10^−12^–5.40 × 10^−55^
** *Cancer* **	82	1.52 × 10^−12^–2.51 × 10^−38^
** *Reproductive System Disease* **	79	1.88 × 10^−13^–1.65 × 10^−31^
** *Hematological Disease* **	70	1.37 × 10^−12^–6.21 × 10^−31^
** *Tumor Morphology* **	44	1.30 × 10^−12^–5.91 × 10^−38^

**Molecular and Cellular Functions**	# Molecules	*p*-value Range
** *Cell Death and Survival* **	72	1.37 × 10^−12^–6.07 × 10^−61^
** *Cellular Development* **	69	8.57 × 10^−13^–2.26 × 10^−35^
** *Cellular Growth and Proliferation* **	69	8.57 × 10^−13^–2.26 × 10^−35^
** *Cell Function and Maintenance* **	52	1.15 × 10^−13^–1.47 × 10^−32^
** *DNA Replication, Recombination, and Repair* **	29	3.41 × 10^−13^–8.15 × 10^−28^

**Physiological System Development and Function**	# Molecules	*p*-value
** *Tissue Development* **	62	8.57 × 10^−13^–2.73 × 10^−33^
** *Organismal Survival* **	44	2.21 × 10^−22^–7.36 × 10^−29^
** *Cardiovascular System Development and Function* **	43	9.90 × 10^−13^–2.73 × 10^−33^
** *Organismal Development* **	39	8.57× 10^−13^–2.73 × 10^−33^
** *Connective Tissue Development and Function* **	23	5.77 × 10^−22^–8.95 × 10^−25^

* The data were generated using Ingenuity Pathway Analysis (IPA) from QIAGEN, USA, March 2023 Release, and incorporated the relative gene expression levels measured in the patients’ group [21]. The *p*-value was measured to understand the likelihood that the association between a set of molecules in our dataset and a related disease or function is due to random association. The smaller the *p*-value (which means a larger −log of that value), the less likely that the association is random and the more significant the association. In general, *p*-values ≤ 0.05 (−log = 1.3) indicate a statistically significant, non-random association. The *p*-value was calculated using a right-tailed Fisher’s exact test [21]. # Molecules: number of significant molecules that are associated with each function.

## Data Availability

The data supporting the findings of this study are available from the corresponding author upon reasonable request.

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
