# Peer review of "Transcriptomics of MASLD Pathobiology in African American Patients in the Washington DC Area †"

_ijms, 2023, doi:10.3390/ijms242316654_

Round 1

Reviewer 1 Report

Comments and Suggestions for Authors

This is a cross-sectional study of African American MASLD patients (hepatic steatosis, metabolic syndrome feature, no other chronic liver disease, unclear if alcohol was exclusion) compared to controls without liver disease who had blood gene expression analyzed. The overall study included 23 MASLD patients and 16 controls, but an Affymetrix microarray for 24,000 loci was performed on 4 MASLD patients and 4 controls. The remaining participants (19 MASLD, 12 controls) had 96 genes significant in HCC assessed via a TaqMan assay. The authors found several differentially expressed pathways, including fibrogenesis and carcinogenesis. Genes that overlapped between the 2 analyses have biological plausibility in MASLD, including E2F1 (lipogenesis) and TGFB1. 

This is an important topic. We know that MASLD prevalence and progression are different across different racial and ethnic groups. We are just scratching the surface of this topic, however, and it is not known what exactly the roles of health care access, environment, food insecurity, genetics, epigenetics etc play in these differences. As the authors point out, African American patients are vastly underrepresented in all clinical studies, including those of MASLD. The main limitation of this study is small sample size, particularly for the global gene expression analysis. Additionally, use of a small FDR of 0.05 means most genes were significantly differentially expressed. 

Comments for the authors:

1.     Would use MASLD terminology throughout the manuscript given recent multi-society guidelines on nomenclature. Making this more straightforward, in this study, subjects had to have confirmed hepatic steatosis and a metabolic comorbidity, essentially meeting MASLD criteria.  

2.     Page 2, Line 48: 48% is an overestimate for MASLD prevalence in the US. Most experts agree it’s around 25-30%. Would update references there.

3.     Page 2, Line 56: would say “The reported incidence of NAFLD among African Americans…”, agree with authors that African Americans are grossly underrepresented in clinical research (of all types, including on MASLD) and therefore this data is not reliable. 

4.     Table 1: the title “Primary Cohort for global gene expression” is a bit misleading. The global gene expression was only done on 4 MASLD patients and 4 controls. Fine to have Table 1 include the whole cohort, but the authors should additionally break this table down into the 2 analyses (4 MASLD, 4 controls and 19 MASLD, 12 controls).

5.     Table 1: did the authors have data on all alcohol consumption (only occasional use is reported)? I assume yes because a questionnaire was administered. I assume that the NAFLD category excluded heavy alcohol use although this is not specifically stated in the Methods; should clarify this.

6.     Page 2, Line 86 and Page 12, Line 411: FDR of 0.05 is low for an analysis with 24,000 loci. This is supported by the finding of differential expression in > 21,000 loci and 82/96 loci significant for TLDA (although TLDA genes were specific to liver disease so this is more believable). Could consider increasing to 0.10. Would defer to a biostatistician reviewer. 

7.     Page 2, Line 88: does the fold change of -1.5 to 1.5 mean that fold changes between these values were included in the number of significant genes, or that fold change of at least 1.5 was the cutoff? Wording is a little confusing. 

8.     Page 3, Line 122: interesting that estrogen receptor signaling was a significant pathway, as has been reported in other studies. Was data on menopause status available? 

9.     Page 6, Line 203: I am confused about the finding in the TLDA analysis that the top pathway was “hepatocellular carcinoma.” I thought all 96 genes were implicated in HCC per the Methods description. 

10.  Discussion: this is a nice overview of prior literature and potential mechanisms. However, I would contextualize the findings by comparing to studies of patients of other races. What are the key differences when these results are compared to what has been found in Hispanic or White patients? I think the big question is, are the mechanisms similar across racial/ethnic groups, and do we need to think about risk stratification differently in different groups? It would help to center the discussion on the unique feature of this study, namely inclusion of African American patients. 

Reviewer 2 Report

Comments and Suggestions for Authors

Even in the abstract sound wired: "The analysis of differentially expressed genes 28 (n=21,448) revealed that 67% and 33% were significantly (p<0.05) up- and downregulated, respectively.

Over half of the transcriptome is altered?

Figure 1 using a heatmap with 4 vs 4 why did they select those samples...

The whole paper does not give anything new or important to me...

Whoever there is much more work to be done. Mainly they utilize IPA and do some network analysis but thats not enough to publish data. Furthermore they need to put their very small dataset somehow into context of bigger datasets which are available.

Comments on the Quality of English Language

--

Reviewer 3 Report

Comments and Suggestions for Authors

The study presented by Mondal et al, performs a transcriptomic analysis in African American NAFLD patients to reveal signature genes and functional pathways associated to this disease. The authors use blood samples (liquid biopsy) as a surrogate tissue instead of liver samples and they try to support the soundness of the method. I believe this is a relevant objective in order to ease the methodology, performing the analysis through a non-invasive method. The problem with this work is that It is not evident that this is the real aim and one is lost in an information without a focus and that in its present form, has many flaws and the research content seems superficial.

-              To support the soundness of the analysis, the study doesn’t present a comparison between the results of Global Transcriptomics in blood samples and any results obtained in NAFLD liver samples. I am sure that this comparison could be made, at least, using NCBI repository data for example:

https://www.ncbi.nlm.nih.gov/geo/query/acc.cgi?acc=GSE135251

https://www.ncbi.nlm.nih.gov/geo/query/acc.cgi?acc=GSE126848

 -              Gene expression validation is made using a human liver cancer-related gene set from Applied Biosystems. I don’t understand the choice, when the NAFLD patients were all in the first stages of the disease and the ones with advanced fibrosis or cirrhosis were precisely excluded from the study.

 -              It seems that the study is made in African American population to find differences in the gene signatures for this ethnic group that help to stratify patients. However, there is no mention of that in the discussion section. On the other hand, did you really expect to find any differences from European, Hispanic or Asian American populations? If they did, the study lacks a comparison with other ethnic populations.

 -              The discussion is a summary of the results and no a reasoning of why the results support the hypothesis and the aim is accomplished or not. For example, I would like to know how can the results help to stratify NAFLD patients at high or at low risk to develop cancer in the future, as the conclusions point out.

Minor:

-          Figures, in general, have poor resolution.

-          Figure 1 is all in blue and red. Figure legend says that downregulated genes are in green but there is not. It would also be helpful to have different colors for NFALD/no NAFLD and Upregulated/downregulated.

-          Legend for figure 2 is difficult to read and should be repeated in figure 4.

-          Table 1 should include those parameters that define the patients as NAFLD, like elastography values, Insulin resistance, type 2 diabetes, % liver steatosis.

-          There are three repeated words at the end of the abstract : with multiethnic populations

Round 2

Reviewer 2 Report

Comments and Suggestions for Authors

I think the extensive revision performed is sufficient to publish in IJMS highligthing now the aims of the study.

Author Response

Comments and Suggestions for Authors: I think the extensive revision performed is sufficient to publish in IJMS highlighting now the aims of the study.

Response: We appreciate the reviewer for thoroughly reviewing and providing important suggestions that enhanced our manuscript.

Reviewer 3 Report

Comments and Suggestions for Authors

- Authors have improved considerably the study, including new results and focusing the objective.

- I also agree very much with the change from NALFD to MASLD; first to use an updated terminology and second because the patients were not pure NAFLD, as they have an almost significant difference in moderate alcohol consumption. Then, I believe that NASH in line 52 of the corrected version, should be MASH.

 - I am sorry but I have to insist in the quality of figures 2, 3 and 4. They are too small to read and when enlarged, the image and letters appear pixelated. Moreover, figure 4 should have the legend as for figure 2.

Author Response

Comments and Suggestions for Authors

- Authors have improved considerably the study, including new results and focusing the objective.

- I also agree very much with the change from NALFD to MASLD; first to use an updated terminology and second because the patients were not pure NAFLD, as they have an almost significant difference in moderate alcohol consumption. Then, I believe that NASH in line 52 of the corrected version, should be MASH.

Response: We thank the reviewer for the suggestion. We have now corrected it as per the suggestion. (Line 45-46, Page 1-2)

 - I am sorry but I have to insist in the quality of figures 2, 3 and 4. They are too small to read and when enlarged, the image and letters appear pixelated. Moreover, figure 4 should have the legend as for Figure 2.

Response: As per the reviewer’s concern we have uploaded high-resolution figures (Figures 2, 3, and 4) in the revised manuscript and added a legend to Figure 4.
